# Multivariate Conformal Prediction using Optimal Transport

## Abstract

Conformal prediction quantifies the uncertainty of machine learning models by constructing sets of plausible outputs instead of relying on a single prediction, which may not exactly match the ground-truth. This is achieved by evaluating all possible output candidates and selecting the most likely ones by ranking their score functions, which measure how well each candidate aligns with the given input, the prediction model, and past observations. Traditionally, this approach has been limited to univariate score functions, as ranking requires a scalar value to order candidates. The challenge lies in extending ranking to multivariate spaces, where no canonical order exists. To address this, we leverage a natural extension of multivariate score ranking based on optimal transport mappings. Our method offers a principled framework for constructing conformal prediction sets in multidimensional settings, preserving distribution-free coverage guarantees with finite data samples.

## 1. Introduction

Conformal prediction (CP) (Gammerman et al., 1998; Vovk et al., 2005; Shafer & Vovk, 2008) has emerged as a simple framework to quantify the prediction uncertainty of machine learning algorithm without relying on distributional assumptions on the data. For a sequence of observed data, and a new input point,

$$D_n = \{(x_1, y_1), ..., (x_n, y_n)\} \text{ and } x_{n+1},$$

the objective is to construct a set that contains the unobserved response $y_{n+1}$ with a specified confidence level $100(1 - \alpha)\%$. This involves evaluating scores $S(x, y, \hat{y}) \in \mathbb{R}$ such as the prediction error of a model $\hat{y}$, for each observation $(x, y)$ in $D_n$ and ranking these score values. The conformal prediction set for the new input $x_{n+1}$ is the collection of all possible responses $y$ whose score $S(x_{n+1}, y, \hat{y})$ ranks small enough to meet the prescribed confidence threshold, compared to the scores $S(x_i, y_i, \hat{y})$ in the observed data.

CP has undergone tremendous developments in recent years,both (Barber et al., 2023; Park et al., 2024; Tibshirani et al., 2019; Guha et al., 2024), that mirror is increased applicability to challenging settings(Straitouri et al., 2023; Lu et al., 2022). To name a few, it has been applied for designing uncertainty sets in active learning (Ho & Wechsler, 2008), anomaly detection (Laxhammar & Falkman, 2015; Bates et al., 2021), few-shot learning (Fisch et al., 2021), time series (Chernozhukov et al., 2018; Xu & Xie, 2021; Chernozhukov et al., 2021; Lin et al., 2022; Zaffran et al., 2022), or to infer the performance guarantee for statistical learning algorithms (Holland, 2020; Cella & Ryan, 2020); and recently to Large Language Models (Kumar et al., 2023; Quach et al., 2023). We refer to the extensive reviews in (Balasubramanian et al., 2014) for other applications to machine learning.

By design, CP requires the notion of order, as the inclusion of a candidate response depends on its relative ranking to the scores observed previously. Hence, the classical strategies developed so far largely targets score functions with univariate outputs. This limits their applicability to multivariate responses, as ranking vector-valued scores $S(x, y, \hat{y}) \in \mathbb{R}^d, d \geq 2$ is evidently not as straightforward as ranking numbers.

**Ordering Vector Distributions using Optimal Transport.** In parallel to these developments, and starting with the seminal reference of (Chernozhukov et al., 2017) and more generally the pioneering works of (Hallin et al., 2021; 2022; 2023), multiple references have explored the possibilities offered by optimal transport theory to define a meaningful ranking or ordering in a multidimensional space. Simply put, the analogous of a rank function computed on data can be found in the optimal Brenier map that transports the data measure to a uniform, symmetric, centered measure of reference in $\mathbb{R}^d$. As a result, a simple notion of univariate rank for a vector $z \in \mathbb{R}^d$ can be found by evaluating the distance of the image of $z$ according to that optimal map to the origin. This approach ensures that the ordering respects both the geometry i.e spatial arrangement of the data and its distribution: points closer to the center get lower ranks.

[1]Anonymous Institution, Anonymous City, Anonymous Region, Anonymous Country. Correspondence to: Anonymous Author <anon.email@domain.com>.

Preliminary work. Under review by the International Conference on Machine Learning (ICML). Do not distribute.

**Contributions** We propose to leverage recent advances in computational optimal transport (Peyré & Cuturi, 2019), using notably differentiable transport map estimators (Pooladian & Niles-Weed, 2021; Cuturi et al., 2019), to leverage the application of such maps in the definition of multivariate score functions. More precisely:

- **OT-CP**: We extend conformal prediction techniques to multivariate score function by leveraging optimal transport ordering, which offers a principled way to define and compute a higher-dimensional quantile and cumulative distribution function. As a result, we obtain distribution-free uncertainty sets that capture the joint behavior of multivariate predictions that enhance the flexibility and scope of conformal predictions.
- We propose a computational approach to this theoretical ansatz using the entropic map (Pooladian & Niles-Weed, 2021) computed from solutions to the Sinkhorn problem (Cuturi, 2013). We prove that our approach preserves the coverage guarantee while being tractable.
- We showcase the application of **OT-CP** using a recently released benchmark of regression tasks (Dheur et al., 2025).

## 2. Background

**Notation** We define $[n] = \{1, \dots, n\}$. We denote the standard uniform measure on $[a, b]$ as $\mathbb{U}([a, b])$. For a discrete set of points $(z_i)_{i \in [n]}$, the empirical uniform measure is denoted $\mathbb{U}_n = \frac{1}{n} \sum_{i=1}^{n} \delta_{z_i}$.

### 2.1. Univariate Conformal Prediction

For a real valued random variable $Z$, it is common to construct an interval $[a, b]$ within which it is expected to fall as

$$\mathcal{R}_\alpha = \{z \in \mathbb{R} : F(z) \in [a, b]\}$$

This is based on the probability integral transform that states that the cumulative distribution function $F$ maps variables to uniform distribution i.e. $\mathbb{P}(F(Z) \in [a, b]) = \mathbb{U}([a, b])$. To guarantee a $(1 - \alpha)$ uncertainty region, it suffices to choose $a$ and $b$ such that $\mathbb{U}([a, b]) \geq 1 - \alpha$ which implies

$$\mathbb{P}(Z \in \mathcal{R}_\alpha) \geq 1 - \alpha. \quad (1)$$

Applying it to the real valued score $Z = S(X, Y)$ of the prediction model $\hat{y}$, an uncertainty set for the response of a given a input $X$ can be expressed as

$$\mathcal{R}_\alpha(X) = \{y \in \mathcal{Y} : F \circ S(X, y) \in [a, b]\}. \quad (2)$$

However, this result is typically not directly usable since the ground-truth distribution $F$ is unknown and must be approximated empirically with $F_n$ using a finite sample of data. When the sample size goes to infinity, one expects to recover Equation (1). The following result provides the tool to obtain the finite sample version.

**Lemma 2.1.** *If $Z_1, \dots, Z_n, Z$ be a sequence of real valued exchangeable random variables, then it holds*

$$F_n(Z) \sim \mathbb{U}\left\{0, \frac{1}{n}, \frac{2}{n}, \dots, 1\right\}$$

$$\mathbb{P}(F_n(Z) \in [a, b]) = \mathbb{U}_{n+1}([a, b]) = \frac{\lfloor nb \rfloor - \lceil na \rceil + 1}{n + 1}.$$

By choosing any $a, b$ such that $\mathbb{U}_{n+1}([a, b]) \geq 1 - \alpha$, Lemma 2.1 guarantees a coverage, that is at least equal to the prescribed level of uncertainty

$$\mathbb{P}(Z \in \mathcal{R}_{\alpha,n}) \geq 1 - \alpha.$$

where, the uncertainty set $\mathcal{R}_{\alpha,n} = \mathcal{R}_\alpha(D_n)$ is defined based on observations $D_n = \{Z_1, \dots, Z_n\}$ and defined as:

$$\mathcal{R}_{\alpha,n} = \{z \in \mathbb{R} : F_n(z) \in [a, b]\}. \quad (3)$$

In short, Equation (3) is an empirical version of Equation (2) based on finite sample data. The striking property is that it preserves the coverage probability $(1 - \alpha)$ and does not depend on the ground-truth distribution of the data.

Given data $D_n$, a prediction model $\hat{y}$ and a new input $X_{n+1}$, one can build an uncertainty set for the unobserved output $Y_{n+1}$ by applying it to observed score functions.

**Proposition 2.2** (Conformal Prediction Coverage)**.** *Consider $Z_i = S(X_i, Y_i)$ for $i$ in $[n]$ and $Z = S(X_{n+1}, Y_{n+1})$ in Lemma 2.1. The conformal prediction set is defined as*

$$\mathcal{R}_{\alpha,n}(X_{n+1}) = \{y \in \mathcal{Y} : F_n \circ S(X_{n+1}, y) \in [a, b]\}$$

*and satisfies a finite sample coverage guarantee*

$$\mathbb{P}(y_{n+1} \in \mathcal{R}_{\alpha,n}(X_{n+1})) \geq 1 - \alpha.$$

The surprising facts are that the coverage guarantee in Proposition 2.2, holds for the *unknown* ground-truth distribution of the data $\mathbb{P}$, does not require quantifying the estimation error $|F_n - F|$, and is applicable to any prediction model $\hat{y}$ as long as it treats the data exchangeably, e.g., a pre-trained model independent of $D_n$.

Leveraging the quantile function $F_n^{-1} = Q_n$, and by setting $a = 0$ and $b = 1 - \alpha$, we have the usual description

$$\mathcal{R}_{\alpha,n}(X_{n+1}) = \{y \in \mathcal{Y} : S(X_{n+1}, y) \leq Q_n(1 - \alpha)\}$$

namely the set of all possible responses whose score rank is smaller or equal to $\lceil (1 - \alpha)(n + 1) \rceil$ compared to ranking of previously observed scores. For the absolute value difference score function, the CP set corresponds to

$$\mathcal{R}_{\alpha,n}(X_{n+1}) = [\hat{y}(X_{n+1}) \pm Q_n(1 - \alpha)].$$

**Center-Outward View** Another classical choice is $a = \frac{\alpha}{2}$ and $b = 1 - \frac{\alpha}{2}$. In that case, we have the usual confidence set that corresponds to the range of values that captures the central proportion with $\alpha/2$ of the data lying below $Q(\alpha/2)$ and $\alpha/2$ lying above $Q(1 - \alpha/2)$.

Introducing the center-outward distribution of $Z$ as the function $T = 2F - 1$, the probability integral transform $T(Z)$ is uniform in the unit ball $[-1, 1]$. This ensures a symmetric description of $\mathcal{R}_\alpha = T^{-1}(B(0, 1 - \alpha))$ around a central point such as the median $Q(1/2) = T^{-1}(0)$. and the radius of the ball now directly corresponds to the desired confidence level of uncertainty. Similarly, we have the empirical center outward distribution $T_n = 2F_n - 1$ and the center-outward view of the conformal prediction set follows as

$$\mathcal{R}_{\alpha,n}(X_{n+1}) = \{y \in \mathcal{Y} : |T_n \circ S(X_{n+1}, y)| \leq 1 - \alpha\}.$$

If $Z$ follows a probability distribution $\mathbb{P}$, then the transformation $z \mapsto T(z)$ is mapping the source distribution $\mathbb{P}$ to the uniform distribution $\mathbb{U}$ over the unit ball. In fact, it can be characterized as essentially the unique monotone increasing function such that $T(Z)$ is uniformly distributed.

**2.2. Multivariate Conformal Prediction**

As recalled in (Dheur et al., 2025), several alternative conformal prediction approaches have been proposed to tackle multivariate prediction problems. While many conformal methods exist for univariate prediction, we focus here on those applicable to *multivariate* outputs. Some of these methods can directly operate using a simple predictor (e.g., a conditional mean) of the response $y$, while some may require stronger assumptions, such as requiring an estimator of the *joint* probability density function between $x$ and $y$, or access to a generative model that mimics the *conditional* distribution of $y$ given $x$) (Izbicki et al., 2022; Wang et al., 2022).

We restrict our attention in this work to approaches that make no such assumption, reflecting our modeling choices for **OT-CP**.

**M-CP**. We will consider the template approach of (Zhou et al., 2024) to use classical CP by aggregating a score function computed on each of the $d$ outputs of the multivariate response. Given a conformity score $s_i$ (to be defined next) for the $i$-th dimension, Zhou et al. (2024) define the following aggregation rule:

$$s_{\text{M-CP}}(x, y) = \max_{i \in [d]} s_i(x, y_i). \tag{4}$$

As (Dheur et al., 2025), we will use *conformalized quantile regression* (Romano et al., 2019) to define the score functions above, for each output $i \in [d]$, where the conformity score is given by:

$$s_i(x, y_i) = \max\{\hat{l}_i(x) - y_i, y_i - \hat{u}_i(x)\},$$

with $\hat{l}_i(x)$ and $\hat{u}_i(x)$ representing the lower and upper conditional quantiles of $Y_i | X = x$ at levels $\alpha_l$ and $\alpha_u$, respectively. In our experiments, we consider equal-tailed prediction intervals, where $\alpha_l = \frac{\alpha}{2}$, $\alpha_u = 1 - \frac{\alpha}{2}$, and $\alpha$ denotes the miscoverage level.

**Merge-CP**. An alternative approach is simply to use a squared Euclidean aggregation,

$$s(x, y) := \|\hat{y}(x) - y\|_2,$$

where the choice of the norm (e.g., $\ell_1$, $\ell_2$, or $\ell_\infty$) depends on the desired sensitivity to errors across tasks. This approach reduces the multidimensional residual to a scalar conformity score, leveraging the natural ordering of the real numbers. This simplification not only makes it straightforward to apply univariate conformal prediction methods, but also avoids the complexities of directly managing vector-valued scores in conformal prediction. A variant consists of applying a Mahalanobis norm (Johnstone & Cox, 2021) in lieu of the squared Euclidean norm, using the covariance matrix $\Sigma$ estimated from the training data (Johnstone & Cox, 2021; Katsios & Papadopulos, 2024),

$$s(x, y) := \|\Sigma^{-1/2}(\hat{y}(x) - y)\|_2,$$

**2.3. Kantorovich Ranks**

A naive way to define ranks in multiple dimensions might be to measure how far each point is from the origin and then rank them by that distance. This breaks down if the distribution of the data is stretched or skewed in certain directions. To correct for this, Hallin et al. (2021) developed a formal framework of center-outward distributions and quantiles, also called Kantorovich ranks (Chernozhukov et al., 2017), extending the familiar univariate concepts of ranks and quantiles into higher dimensions, building on elements of optimal transport theory.

Let $\mu$ and $\nu$ be source and target probability measures on $\Omega \subset \mathbb{R}^d$. We consider the optimal transport problem with square Euclidean cost

$$\inf_{\pi \in \Pi(\mu, \nu)} \int_{\Omega \times \Omega} \|\boldsymbol{x} - \boldsymbol{y}\|^2 \, d\pi(\boldsymbol{x}, \boldsymbol{y}),$$

where $\Pi(\mu, \nu)$ is the set of all transport plans, i.e. joint distributions $\pi$ on $\Omega \times \Omega$ whose marginals are $\mu$ and $\nu$.

**Optimal Transport Map** One can look for a map $T : \Omega \to \Omega$ that pushes forward $\mu$ to $\nu$ and minimizes the average transportation cost

$$T^\star \in \arg\min_{T_\# \mu = \nu} \int_\Omega \|x - T(x)\|^2 \, d\mu(x). \tag{5}$$

Brenier's theorem states that if the source measure $\mu$ has a density, there exists a solution to 5 that is the gradient of a convex function $\phi : \Omega \to \mathbb{R}$ such that $T^\star = \nabla \phi$.

In the one-dimensional case, the cumulative distribution function of a distribution $\mathbb{P}$ is the unique increasing function transporting it to the uniform distribution. This monotonicity property generalizes to higher dimensions through the gradient of a convex function $\nabla\phi$. Thus, one may view the optimal transport map in higher dimensions as a natural analog of the univariate cumulative distribution function both represent the unique monotone way to send one probability distribution onto another.

**Definition 2.3.** The center-outward distribution of a random variable $Z \sim \mathbb{P}$ is defined as the optimal transport map $T = \nabla\phi$ that pushes $\mathbb{P}$ forward to the uniform distribution $\mathbb{U}$ on the unit ball $B(0,1)$. The rank of $Z$ is defined as $\text{Rank}(Z) = \|T(Z)\|$, the distance to origin.

**Quantile region** is an extension of quantiles to multiple dimensions to represent region in the sample space that contains a given proportion of probability mass. The quantile region at probability level $(1 - \tau) \in (0,1)$ can be defined as

$$\mathcal{R}_\tau = \{z \in \mathbb{R}^d : \|T(z)\| \leq 1 - \tau\}.$$

By definition of the spherical uniform distribution, we have $\|T(Z)\|$ is uniform on $(0,1)$ which implies

$$\mathbb{P}(Z \in \mathcal{R}_\tau) = 1 - \tau. \tag{6}$$

## 3. Kantorovich Conformal Prediction

### 3.1. Multi-Output Conformal Prediction

We consider that $\mathbb{P}$ is only available through a finite set of samples $\{Z_i\}_{i=1}^{n+1}$ and a grid of $\mathbb{U}$ with as many points. We consider first the *discrete* transport map

$$T_{n+1} : \{Z_i\}_{i=1}^{n+1} \to \{U_i\}_{i=1}^{n+1}$$

which can be obtained by solving the optimal assignment problem, which seeks to minimize the total transport cost between the empirical distributions $\mathbb{P}_{n+1}$ and $\mathbb{U}_{n+1}$:

$$T_{n+1} \in \arg\min_{T \in \mathcal{T}} \sum_{i=1}^{n+1} \|Z_i - T(Z_i)\|^2,$$

where $\mathcal{T}$ is the set of bijections mapping the observed sample $\{Z_i\}_{i=1}^{n+1}$ to the grid $\{U_i\}_{i=1}^{n+1}$.

**Definition 3.1.** Let $(Z_1, \ldots, Z_n, Z_{n+1})$ be a sequence of exchangeable variables in $\mathbb{R}^d$ that follow a common distribution $\mathbb{P}$. The discrete center-outward distribution $T_{n+1}$ is the transport map pushing forward $\mathbb{P}_{n+1}$ to $\mathbb{U}_{n+1}$.

When dealing with empirical distribution with finite samples $Z_1, \ldots, Z_n, Z_{n+1}$ In this asymptotic regime (Chewi et al., 2024), the empirical source distribution $\mathbb{P}_{n+1}$ approximates the ground-truth $\mathbb{P}$ and as well as the empirical transport map $T_{n+1}$ approximates in sample the exact transport $T^\star$.

Following (Hallin et al., 2021) to formalize the discrete spherical uniform distribution and its associated empirical cumulative distribution function, we begin by stating the construction of the discrete spherical uniform distribution involves a uniform grid defined such that the total number of points $n = n_R n_S + n_o$, where $n_o$ points are at the origin.

- $n_S$ unit vectors $\mathbf{u}_1, \ldots, \mathbf{u}_{n_S}$ are uniform on the sphere.

- $n_R$ radius are regularly spaced as $\left\{\frac{1}{n_R}, \frac{2}{n_R}, \ldots, 1\right\}$.

The grid discretizes the sphere into layers of concentric shells, with each shell containing $n_S$ equally spaced points along directions determined by the unit vectors. The discrete spherical uniform distribution puts equal mass over each points of the grid that is to say $n_o \times 1/n$ mass on the origin and $1/n$ on the remaining. This ensures isotropic sampling at fixed radius onto $[0,1]$.

By definition of the target distribution $\mathbb{U}_{n+1}$, it holds

$$\|T_{n+1}(Z_{n+1})\| \sim \mathbb{U}\left\{0, \frac{1}{n_R}, \frac{2}{n_R}, \ldots, 1\right\}.$$

In order to define an empirical quantile region as Equation (6), we need an extrapolation $\bar{T}_{n+1}$ of $T_{n+1}$ out of the samples $(Z_i)_{i \in [n+1]}$. By definition of such maps

$$\|\bar{T}_{n+1}(Z_{n+1})\| = \|T_{n+1}(Z_{n+1})\|$$

is still uniformly distributed and the empirical quantile region can be defined as

$$\mathcal{R}_{\alpha,n+1} = \{z \in \mathbb{R}^d : \|\bar{T}_{n+1}(z)\| \leq 1 - \alpha\}$$

and expect that $\mathbb{P}(Z \in \mathcal{R}_{\alpha,n+1}) \approx 1 - \alpha$ when $n$ is large.

Nevertheless, the core point of conformal prediction methodology is to go beyond asymptotic results or regularity assumptions about the data distribution. This is crucial because we only have access to a finite amount of data, and the ground-truth distribution of the data is unknown in practice. In that case, it is not immediate to have guarantee with respect to the ground-truth distribution such as Equation (7).

### 3.2. Optimal Transport Merging

We introduce the Optimal Transport Merging, a simple procedure that reduces any vector-valued score $S(x,y) \in \mathbb{R}^d$ in a one-dimension score. More precisely, we define the new non-conformity score function of an observation as

$$S_{\text{OT-CP}}(x,y) = \|T^\star \circ S(x,y)\|_2$$

where $T^\star$ is the optimal Brenier (1991) map that pushes the distribution of vector-valued scores onto the uniform

ball distribution $\mathbb{U}$ in the same approach. This approach allows us to exploit the natural ordering of the real line, making it possible to directly apply one-dimensional conformal prediction methods to the sequence of transformed scores $Z_i = \|S_{\mathrm{OT-CP}}(X_i, Y_i)\|_2$ for $i \in [n+1]$.

In practical implementation, $T^\star$ can be replaced by any approximation $\hat{T}$ that preserves the permutation invariance of the score functions. We introduce the conformal prediction set resulting from the optimal transport merging is

$$\mathcal{R}_{\mathrm{OT-CP}}(X_{n+1}, \alpha) = \mathcal{R}_\alpha(T, X_{n+1})$$

with respect to a given transport map $T$

$$\mathcal{R}_\alpha(T) = \left\{ y : F_n(\|S_{\mathrm{OT-CP}}(X_{n+1}, y)\|_2) \leq 1 - \alpha \right\}.$$

have a coverage $(1 - \alpha)$, where $F_n$ is empirical (univariate) cumulative distribution function of the observed scores

$$\left\{ \|S_{\mathrm{OT-CP}}(X_1, Y_1)\|, \ldots, \|S_{\mathrm{OT-CP}}(X_n, Y_n)\| \right\}.$$

Proposition 2.2 implies

$$\mathbb{P}(Y_{n+1} \in \mathcal{R}_{\mathrm{OT-CP}}(X_{n+1})) \geq 1 - \alpha.$$

*Remark* 3.2. Our proposed conformal prediction framework **OT-CP** with optimal transport merging score function generalizes the **Merge-CP** approaches. More specifically, under the additional assumption that we are transporting a source Gaussian (resp. uniform) distribution to a target Gaussian (resp. uniform) distribution, the transport map is linear (Peyré & Cuturi, 2019; Muzellec & Cuturi, 2018)

### 3.3. Coverage Guarantees under Approximations

When dealing with high-dimensional data or complex distributions, it is essential to find computationally feasible methods to approximate the optimal transport map $T^\star$ with a map $\hat{T}$. In practical applications, we will rely on empirical approximations of the Brenier (1991) map using finite samples. Note that this approach may encouter a few statistical roadblocks, as such estimators are significantly hindered by the curse of dimensionality (Chewi et al., 2024). Consequently, one may think that these maps, not serving as reliable approximations, may hurt the performance of our approach. However, the machinery of conformal prediction presented earlier in the background section allows to maintain a coverage level, irrespective of sample size limitations. We defer the presentation of this practical approach to section 3.4 and focus first on coverage guarantees.

**Coverage of Approximated Quantile Region**
Let us assume an arbitrary approximation $\hat{T}$ of the Brenier (1991) map and define the corresponding quantile region as

$$\mathcal{R}(\hat{T}, r) = \{ z \in \mathbb{R}^d : \|\hat{T}(z)\| \leq r \},$$

The coverage in Equation (7) is not automatically maintained since $\hat{\mathbb{U}} := \hat{T}_{\#}\mathbb{P}$ may not coincide with $\mathbb{U}$. As a result, the validity of the approximated quantile region may be compromised unless we can control the magnitude of the error $\|\hat{\mathbb{U}} - \mathbb{U}\|$, which requires additional regularity assumptions.

In its standard formulation, conformal prediction relies on an empirical setting and does not directly apply to the continuous case. Consequently, it does not provide a solution for calibrating entropic quantile regions, for example. However, a careful inspection of the one-dimensional case reveals that understanding the distribution of the probability integral transform is the key point:

- $\mathbb{U}\left(\left\{0, \frac{1}{n}, \frac{1}{2}, \ldots, 1\right\}\right) \sim F_n(Z) \neq F(Z) \sim \mathbb{U}(0, 1)$ .

Instead of relying on an analysis of approximation error to quantify the deviation $|F_n - F|$ under certain regularity conditions, conformal prediction fully characterizes the distribution of the probability integral transform and calibrates the radius of the quantile region accordingly.

We follow this very simple idea and note that by definition

$$\mathbb{P}(\mathcal{R}(\hat{T}, r)) = \mathbb{P}(\|\hat{T}(z)\| \leq r) = \hat{\mathbb{U}}(B(0, r)).$$

Instead of relying on $\hat{\mathbb{U}} \approx \mathbb{U}$, we define

$$r_\alpha(\hat{T}, \mathbb{P}) = \inf\{r : \hat{\mathbb{U}}(B(0, r)) \geq 1 - \alpha\}$$

that leads to a desired coverage with the approximated transported map . For a radius $\hat{r}_\alpha = r_\alpha(\hat{T}, \mathbb{P})$, it holds

$$\mathbb{P}\left(Z \in \mathcal{R}(\hat{T}, \hat{r}_\alpha)\right) \geq 1 - \alpha.$$

By extension, a quantile region of the vector-valued score $Z = S(X, Y) \in \mathbb{R}^d$ of a prediction model $\hat{y}$ provides an uncertainty set for the response of a given a input $X$, with prescribed coverage $(1 - \alpha)$ can be expressed as

$$\mathcal{R}_\alpha(X) = \left\{ y \in \mathcal{Y} : \|T \circ S(X, y)\| \leq 1 - \alpha \right\}.$$

$$\mathbb{P}(Y \in \mathcal{R}_\alpha(X)) = 1 - \alpha. \tag{7}$$

In the following result, we give the finite sample analog of Equation (6), which provides a finite sample guarantee for our optimal transport approach.

**Lemma 3.3** (Coverage of Empirical Quantile Region). *Let $Z_1, \ldots, Z_n, Z_{n+1}$ be a sequence of exchangeable variable in $\mathbb{R}^d$, then, $\mathbb{P}(Z_{n+1} \in \mathcal{R}_{\alpha, n+1}) \geq 1 - \alpha$.*

Remark that the source probability in Lemma 3.3 is the ground-truth $\mathbb{P}$. Given a transport map $\hat{T}$ and applying and the empirical radius $r_{\alpha, n+1} = r_\alpha(\hat{T}, \mathbb{P}_{n+1})$, it holds

$$\mathbb{P}_{n+1}(Z_{n+1} \in \mathcal{R}(\hat{T}, r_{\alpha, n+1})) \geq 1 - \alpha.$$

However, this is *only* an empirical coverage statement:

$$\frac{1}{n+1} \sum_{i=1}^{n+1} \mathbb{1}\{Z_i \in \mathcal{R}(\hat{T}, r_{\alpha,n+1})\} \geq 1 - \alpha$$

which does not implies coverage with respect to $\mathbb{P}$ unless $n \to \infty$. The following steps show how to obtain finite sample validity.

*Proof.* For simplicity, we will denote the quantile region as $\mathcal{R}_{\alpha,n+1} = \mathcal{R}(\hat{T}, r_{\alpha,n+1})$. Then by exchangeability:

$$\mathbb{P}(Z_{n+1} \in \mathcal{R}_{\alpha,n+1}) = \frac{1}{n+1} \sum_{i=1}^{n+1} \mathbb{P}_{n+1}(Z_i \in \mathcal{R}_{\alpha,n+1})$$

$$= \mathbb{E}\left[\frac{1}{n+1} \sum_{i=1}^{n+1} \mathbb{1}\{Z_i \in \mathcal{R}_{\alpha,n+1}\}\right]$$

$$= \mathbb{E}\left[\mathbb{P}_{n+1}(Z_{n+1} \in \mathcal{R}_{\alpha,n+1})\right]$$

$$\geq 1 - \alpha.$$

$\square$

This can be directly applied to obtain conformal prediction set for vector-valued non-conformity score functions $Z_i = S(X_i, Y_i) \in \mathbb{R}^d$ for $i$ in $[n+1]$ in Lemma 3.3.

**Proposition 3.4.** *The conformal prediction set is defined as*

$$\widehat{\mathcal{R}}_{\alpha,n+1}(X_{n+1}) = \left\{ y \in \mathcal{Y} : \|\hat{T} \circ S(X_{n+1}, y)\| \leq \hat{r}_{\alpha,n+1} \right\}$$

*with $\hat{r}_{\alpha,n} = \inf\left\{r \geq 0 : \hat{\mathbb{U}}(B(0,r)) \geq 1 - \alpha\right\}$. It satisfies a distribution-free finite sample coverage guarantee*

$$\mathbb{P}\left(Y_{n+1} \in \widehat{\mathcal{R}}_{\alpha,n+1}(X_{n+1})\right) \geq 1 - \alpha. \qquad (8)$$

Approaches relying on vector-valued probability integral transform, e.g., leveraging Copulas have been explored recently (Messoudi et al., 2021; Park et al., 2024) and concluded that loss of coverage can occur when the estimated copula of the scores deviates from the true copula and thus does not formally guarantee finite-sample validity. To our knowledge, Proposition 3.4 provides the first calibration guarantee for such confidence regions without assumptions on the distribution, for any approximation map $\hat{T}$. Specifically using the discrete spherical uniform grid implies:

**Proposition 3.5.** *Given $n$ discrete sample points distributed over a sphere with radius $\{0, \frac{1}{n_R}, \frac{2}{n_R}, \ldots, 1\}$ and directions uniformly sampled on the sphere, the smallest radius to obtain a coverage $(1 - \alpha)$ is determined by*

$$r_\alpha = \frac{j_\alpha}{n_R} \text{ where } j_\alpha = \left\lceil \frac{n(1-\alpha) - n_o}{n_S} \right\rceil,$$

*where $n_S$ is the number of directions, $n_R$ is the number of radius, and $n_o$ is the number of copies of the origin.*

*Remark* 3.6. When the discrete transport problem is solved approximately and one obtain $\hat{T}_{n+1}$, then choosing $\hat{r}_{\alpha,n+1} = r_\alpha(\hat{T}_{n+1}, \mathbb{P}_{n+1})$ ensure finite sample coverage just as Section 3.3. So one can take benefit of numerical efficiency without sacrificing valid coverage.

### 3.4. Implementation with the Entropic Map

We assume access to two sample sets, i.e., one containing residuals $\hat{\mu}_n = \frac{1}{n}\sum_i \delta_{z^i}$, and the second containing the discretized uniform grid on the sphere, $\hat{\nu}_m = \frac{1}{m}\sum_j \delta_{u^j}$, not necessarily assuming a same size, namely $n \neq m$. A convenient estimator for the Brenier map $T^\star$ is the entropic map (Pooladian & Niles-Weed, 2021). Let $\varepsilon > 0$ and write $K_{ij} = [\exp(-\|z^i - u^j\|^2/\varepsilon)]_{ij}$ the kernel matrix. One can then define,

$$\mathbf{f}^\star, \mathbf{g}^\star = \underset{\mathbf{f}\in\mathbb{R}^n, \mathbf{g}\in\mathbb{R}^m}{\operatorname{argmax}} \langle \mathbf{f}, \tfrac{1_n}{n} \rangle + \langle \mathbf{g}, \tfrac{1_m}{m} \rangle - \varepsilon \langle e^{\frac{\mathbf{f}}{\varepsilon}}, K e^{\frac{\mathbf{g}}{\varepsilon}} \rangle. \quad (9)$$

Problem (9) is an unconstrained concave optimization problem known as the regularized OT problem in dual form (**?**, Prop. 4.4). Problem (9) can be solved numerically with the Sinkhorn algorithm (Cuturi, 2013). Equipped with these optimal vector, one can define the maps, valid out of sample,

$$f_\varepsilon(z) = \min_\varepsilon([\|z - u^j\|^2 - \mathbf{g}_j^\star]_j), \qquad (10)$$

$$g_\varepsilon(u) = \min_\varepsilon([\|z^i - u\|^2 - \mathbf{f}_i^\star]_i), \qquad (11)$$

where for a vector $\mathbf{u}$ or arbitrary size $s$ we define the log-sum-exp operator as $\min_\varepsilon(\mathbf{u}) := -\varepsilon \log(\frac{1}{s}\mathbf{1}_s^T e^{-\mathbf{u}/\varepsilon})$. Using the Brenier (1991) theorem, linking potential values to optimal map estimation, one obtains an estimator for $T^\star$:

$$T_\varepsilon(z) := z - \nabla f_\varepsilon(z) = \sum_{j=1}^m p^j(z)u^j, \qquad (12)$$

where the weights depend on $z$ as:

$$p^j(z) := \frac{\exp\left(-\left(\|z - u^j\|^2 - \mathbf{g}_j^\star\right)/\varepsilon\right)}{\sum_{k=1}^m \exp\left(-\left(\|z - u^k\|^2 - \mathbf{g}_k^\star\right)/\varepsilon\right)}. \qquad (13)$$

One can obtain, analogously, an estimator for the inverse map $(T^\star)^{-1}$ using the potential $g_\varepsilon$, as demonstrated in Fig. 5. Using the entropic map estimator requires running the Sinkhorn (1964) algorithm on a $n \times m$ cost matrix at train time, and at each evaluation, compute weights in (13) that require computing the distance of any incoming point $z$ to the uniform grid. The complexity is therefore $O(nm)$ when training the map and conformalizing its scores, and then $O(m)$ at each evaluation of a score for a given $y$.

**Sampling on the sphere** As mentioned in (Hallin et al., 2021), it is preferable to sample the uniform measure $\mathbb{U}_d$ with diverse samples, and this can be achieved using stratified sampling on radii lengths, but, most importantly, low-discrepancy samples on the sphere to pick sampling directions. We borrow inspiration from the review provided in

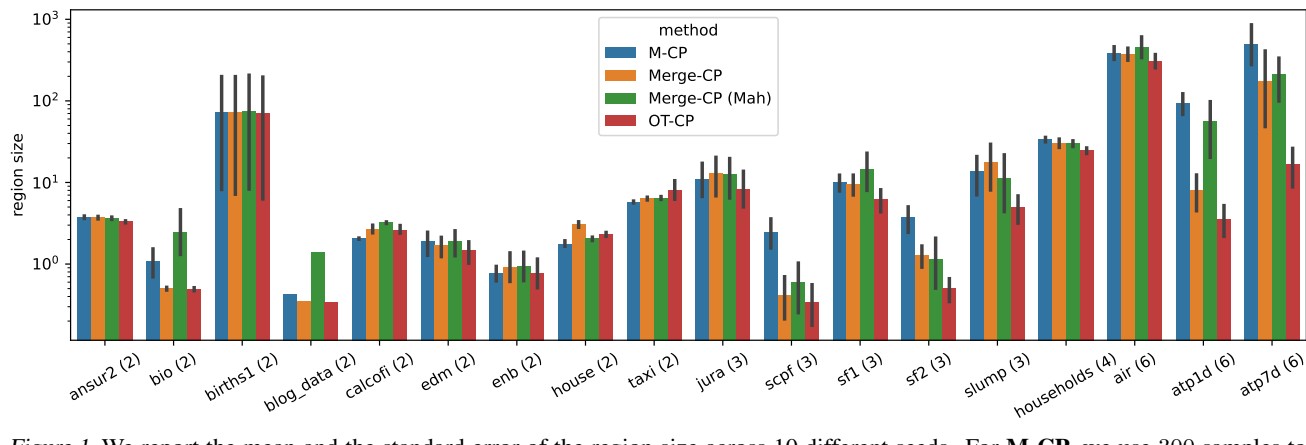

*Figure 1.* We report the mean and the standard error of the region size across 10 different seeds. For **M-CP**, we use 300 samples to compute the conditional mean, and for **OT-CP**, we use $\varepsilon = 0.1$ and $2^{15} = 32768$ points in the uniform target measure. On average, **OT-CP** displays smaller region size than other baselines. The dimensionality of each dataset is provided for reference underneath, datasets are sorted in increasing dimension order.

(Nguyen et al., 2024) to pick their *Gaussian based mapping* approach (Basu, 2016). This consists in mapping a low-discrepancy sequence $w_1, \ldots, w_L$ on $[0, 1]^d$ to a potentially low-discrepancy sequence $\theta_1, \ldots, \theta_L$ on $\mathbb{S}^{d-1}$ through the mapping $\theta = \Phi^{-1}(w)/\|\Phi^{-1}(w)\|_2$, where $\Phi^{-1}$ is the inverse CDF of $\mathcal{N}(0, 1)$ applied entry-wise.

## 4. Experiments

### 4.1. Setup and Metrics

We borrow the experimental setting provided by Dheur et al. (2025) and benchmark multivariate conformal methods on a total of 24 tabular datasets. Total data size $n$ in these datasets ranges from 103 to 50,000, with input dimension $p$ ranging from 1 to 348, and output dimension $d$ ranging from 2 to 16. We adopt their approach, which is to rely on a multivariate quantile function forecaster (MQF[2], Kan et al., 2022), a normalizing flow that is able to quantify output uncertainty conditioned on input $x$. However, in accordance with our stance mentioned in the background section, we will only assume access to the conditional mean (point-wise) estimator for **OT-CP**.

As is common in the field, we evaluate the methods using several metrics, including marginal coverage (MC), and mean region size (Size). The latter is using importance sampling, leveraging (when computing test time metrics only), the generative flexibility provided by the MQF[2] as an invertible flow. See (Dheur et al., 2025) and their code for more details on the experimental setup.

### 4.2. Hyperparameter Choices

We apply default parameters for all three competing methods, **M-CP** and **Merge-CP**, using (or not) the Mahalanobis correction. For **M-CP** using conformalized quantile regres-

sion boxes, we follow (Dheur et al., 2025) and leverage the empirical quantiles return by MQF[2] to compute boxes (Zhou et al., 2024).

**OT-CP** our implementation requires essentially tuning two important hyperparameters: the entropic regularization $\varepsilon$ and the total number of points used to discretize the sphere $m$, not necessarily equal to the input data sample size. These two parameters describe a fundamental statistical and computational trade-off. On the one hand, it is known that increasing $m$ will mechanically improve the ability of $T_\varepsilon$ to recover in the limit $T^\star$ (or at least solve the semi-discrete (Peyré & Cuturi, 2019) problem of mapping $n$ data points to the sphere). However, large $m$ incurs a heavier computational price when running the Sinkhorn algorithm. On the other hand, increasing $\varepsilon$ improves on *both* computational and statistical aspects, but deviates further the estimated map from the ground truth $T^\star$ to target instead a blurred map. We have experimented with these aspects and derive from our experiments that both $m$ and $\varepsilon$ should be increased to track increase in dimension. As a sidenote, we do observe that debiasing the outputs of the Sinkhorn algorithm does not result in improved results, which agrees with the findings in (Pooladian et al., 2022).

### 4.3. Results

We present results by differentiating datasets with small dimension $d \leq 6$ from datasets with higher dimensionality, that we expect to be more challenging to handle with OT approaches, owing to the curse of dimensionality that might degrade the quality of multivariate quantiles.

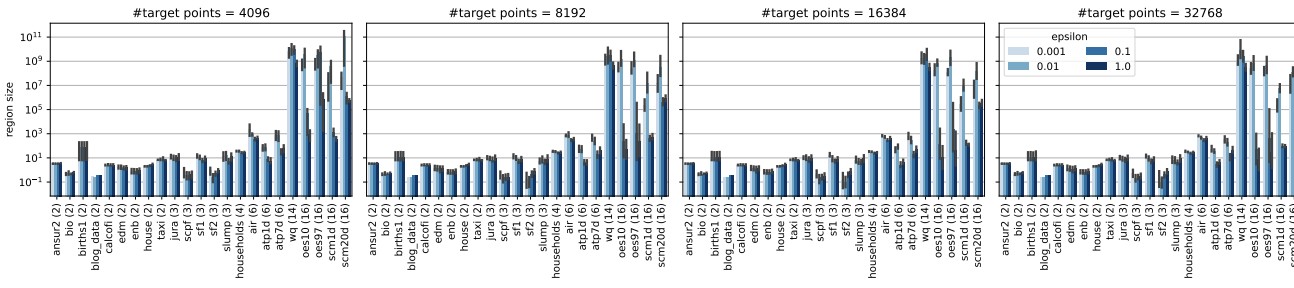

Figure 2. Ablation on both the total number of points $m$ sampled from the sphere and the $\varepsilon$ regularization level for all datasets. This plot details the impact of the two important hyperparameters we single out in **OT-CP**. As can be seen, larger sample size $m$ improves region size (smaller the better) for roughly all datasets and regularizations. On the other hand, one must tune $\varepsilon$ to operate at a suitable regime: not too low, which results in the well documented poor statistical performance of unregularized OT, nor too high, which would lead to a collapse of the entropic map to the sphere.

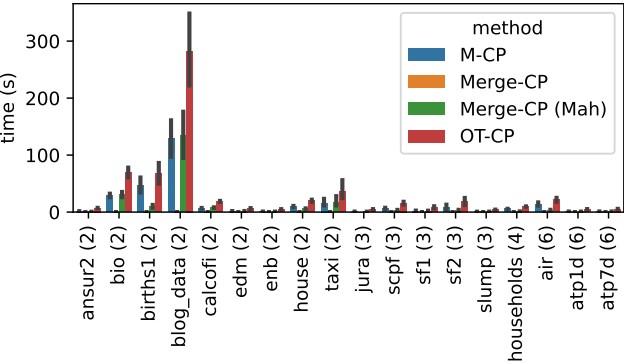

Figure 3. Computational time on small dimensional datasets. **OT-CP** incurs more compute time due to the OT map estimation. See Fig.7 for a similar picture for higher dimensional datasets.

# 5. Conclusion

We have proposed **OT-CP**, a new approach that can leverage a recently proposed formulation for multivariate quantiles that uses optimal transport theory and optimal transport map estimators. We show the theoretical soundness of this approach, but, most importantly, demonstrate its applicability throughout a broad range of tasks compiled by (Dheur et al., 2025). Compared to similar baselines that either leverage a conditional mean regression estimator (**Merge-CP**), or more involved quantile regression estimators (**M-CP**), **OT-CP** displays superior performance overall, while incurring, predictably, a higher train / calibration time cost. The challenges brought forward by the estimation of OT maps in high dimensions (Chewi et al., 2024) require being particularly careful when tuning entropic regularization and grid size. However, we show that there exists a reasonable setting for both these parameters that delivers good performance across most tasks.

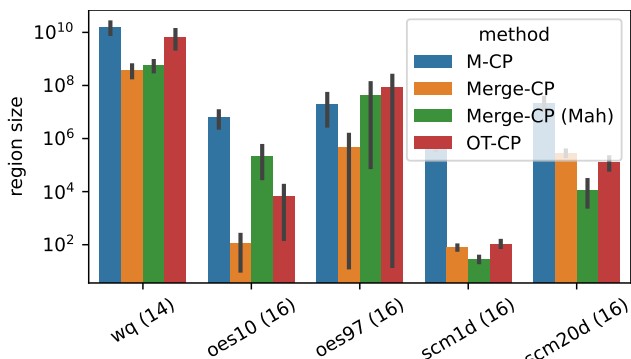

Figure 4. As in 1, we report mean and standard errors for region size across 10 different seeds for larger datasets. We keep the same parameters and importantly $\varepsilon = 0.1$ and $2^{15} = 32768$ points in the uniform target measure.

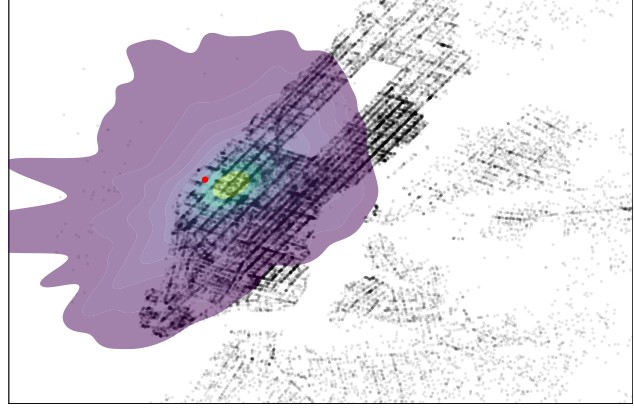

Figure 5. Conformal $\alpha = 5\%$ sets recovered by mapping back the reduced sphere on the Manhattan map, in agreement with Equation 7, on a prediction for the `taxi` dataset. We use the inverse entropic map mentioned in Section 3.4, mapping back the gridded sphere of size $m = 2^{15}$.

## Impact Statement

This paper presents work whose goal is to advance the field of Machine Learning. There are many potential societal consequences of our work, none which we feel must be specifically highlighted here.

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

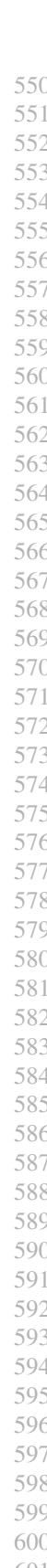

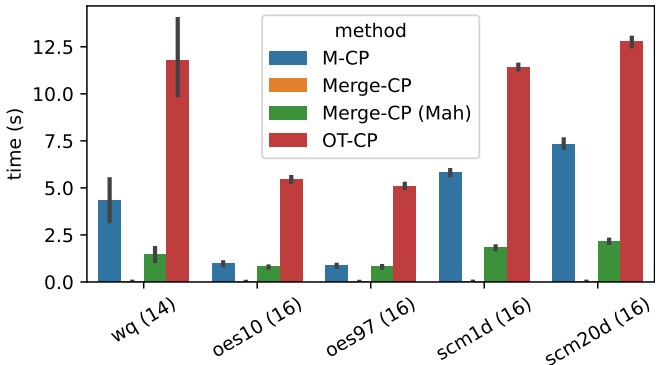

*Figure 6.* Coverage for bigger dimensional datasets, corresponding to the setting displayed in Figure 6

*Figure 7.* Runtimes for bigger dimensional datasets, corresponding to the setting displayed in Figure 6

# A. Appendix

We provide a few additional results related to the experiments proposed in Section 4

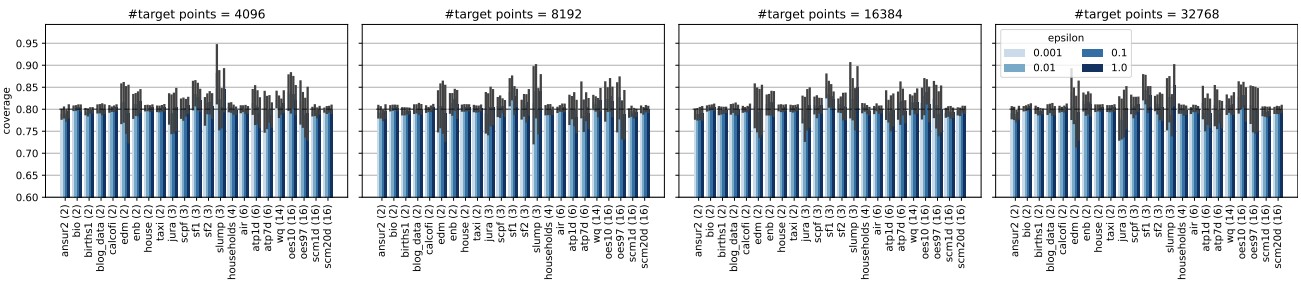

*Figure 8.* Ablation: coverage quality as a function of hyperparameters, with the setting corresponding to Fig.2

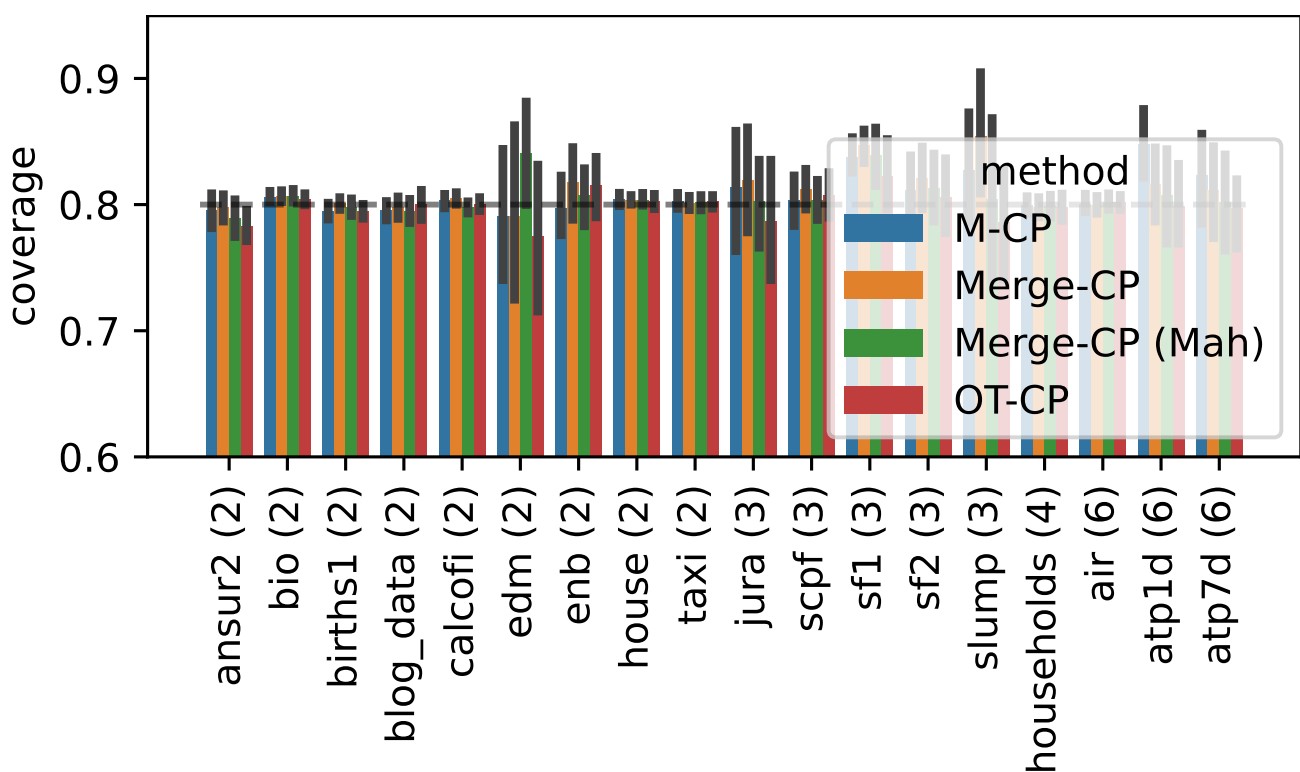

*Figure 9.* Coverage of all baselines on small dimensional datasets, corresponding to the region sizes given in 1.

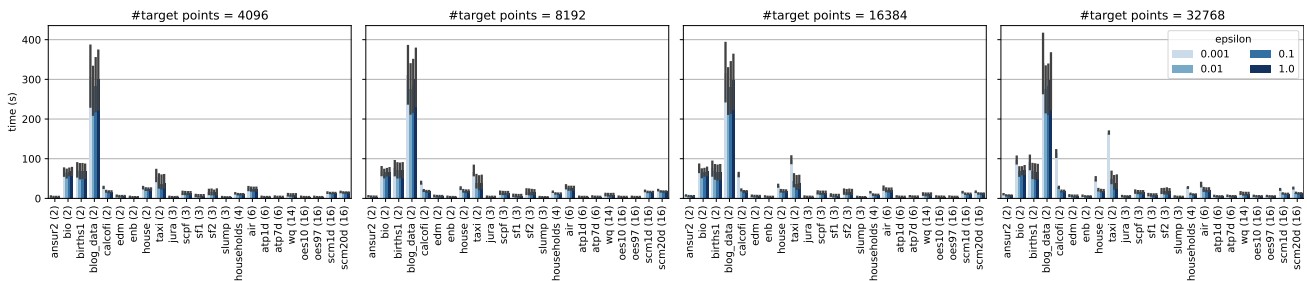

*Figure 10.* Ablation: running time as a function of hyperparameters, with the setting corresponding to Fig.2

