# OpenReview forum: "Multivariate Conformal Prediction using Optimal Transport"
_ICML.cc/2025/Conference — Submitted to ICML 2025_

### Official Review · Reviewer_QqAt · 2025-03-07

**Overall Recommendation:** 2

**Summary:**

This paper proposes a new conformal score function for a multivariate response paired with a Euclidean predictor. The idea behind the score is to use a functional of optimal transport from the d-dimensional score to the uniform distribution. The marginal coverage of the proposed score is guaranteed. The numerical behavior is illustrated with a comparison to other scores on several real-world datasets.

**Claims And Evidence:**

The paper claims that the proposed score can achieve marginal validity and produce relatively small prediction sets. The marginal validity is supported by a proposition, while the latter statement is supported only by numerical results.

**Essential References Not Discussed:**

No essential references not discussed.

**Experimental Designs Or Analyses:**

N/A

**Methods And Evaluation Criteria:**

The proposed new score is evaluated based on the region size and marginal coverage of the prediction sets and also the computing time. I suggest that the author also present the conditional coverage level.

**Other Comments Or Suggestions:**

N/A

**Other Strengths And Weaknesses:**

I have several suggestions that may help improve the next version of this paper:

1. In conformal prediction, conditional coverage is more important than marginal validity in both theoretical and practical analysis. The key contribution of this paper seems to be the new score for multivariate responses, but I believe only marginal validity alone may not be sufficient for publication in a top conference like ICML.

2. The proposed score (16) seems to rely on a pre-selected score function \( S \) for scalar responses. The choice of \( S \) likely has a significant impact on the final results, and I encourage the authors to provide further discussion on this.

3. The numerical results do not demonstrate a clear advantage of the proposed method. The region size in Figure 1 suggests that the proposed score yields comparable results to existing methods on most datasets.

**Questions For Authors:**

N/A

**Relation To Broader Scientific Literature:**

The idea of using optimal transport in vector data inference is not new and was first proposed by Hallin et al. in their 2021 AOS paper. This paper applies that idea to constructing a conformity score for multivariate responses.

**Theoretical Claims:**

This paper only studies the marginal coverage of the prediction sets. Since the proof is standard and of less interest, I did not check its correctness.

---

> ### Author Rebuttal · Authors · 2025-04-01
>
> Many thanks for your detailed review and for the many kind suggestions to improve our work.
>
> >**In conformal prediction, conditional coverage is more important than marginal validity ... but I believe only marginal validity alone may not be sufficient for publication in a top conference like ICML.**
>
> Many thanks for this very constructive point.
>
> First, one should aknowledge that in vector-valued setting marginal validity is not straightforward, in contrast to $d=1$ where the conformity functions can be ranked. These canonical order no longer exists in multiple dimensions and unfortunately, **marginal validity estimation can be lost when these multivariate quantile are not accurately estimated**. These approximation biases can cause failure of marginal coverage even in recent works, for example Semi-parametric Conformal Prediction (J. W. Park etal, 2024)[Section 7: Limitations] which do not preserve marginal validity. In our setting, we additionally leverage entropic map whose approximation errors can also impact the coverage guarantee. This is why our conformalization step (Remark 3.6) is crucial. We will clarify.
>
> We agree with the reviewer that conditional coverage is an important point to consider.
>
> We clarified this on two fronts: metrics and by extending OTCP.
>
> **Results are visible in (anonymized link) https://shorturl.at/alFNM**
>
> * We leverage the implementation of [Dheur et al 25] and report the Worst Slab Coverage (WSC) and CEC-X [Appendix F.6] computed by their pipeline.
>
> * In general, pointwise conditional coverage is impossible to achieve without stronger assumptions on the ground-truth distribution see https://arxiv.org/abs/1203.5422 or https://arxiv.org/abs/1903.04684. We have the same issue here.
> However, following your comment and that of Reviewer **8VoF**, we also propose a simple adaptation of OTCP to approximate conditional coverage by partitioning the features space into regions $\mathcal{X} = \cup_{k=1}^{K} A_k$ and computing a transport map $T_{A_k}$ for every region. Our proof technique directly applies conditional on $A_k$​, and under exchangeability we have  $\mathbb{P}(Y_{n+1} \in \mathcal{R}\_{\alpha}(X_{n+1}) \mid X_{n+1} \in A_k) \geq 1-\alpha$, for every $k \in [K]$. The partition $(A_k)$ are obtained by running $K$-means on the training set. These two baselines are `OTCP-CLS (5)` and `OTCP-CLS (10)`, with $k=5$ and $k=10$, respectively.
>
> >**The proposed score (16) seems to rely on a pre-selected score function ( S ) for scalar responses. The choice of ( S ) likely has a significant impact on the final results, and I encourage the authors to provide further discussion on this.**
>
> We agree, but this is the case for any other vector-valued conformal method. Even in one dimension, the choice of score function depends on the specific task and problem at hand.  In this benchmark, we focused on simple residuals for OTCP to showcase more clearly the benefit of remapping these residues to their quantiles. Our framework covers arbitrary user-specified vector-valued score function.
>
> >**The numerical results do not demonstrate a clear advantage of the proposed method. The region size in Figure 1 suggests that the proposed score yields comparable results to existing methods on most datasets.**
>
> We do not claim dominance indeed, as the breadth of datasets targetted in this benchmark (variety in size, dimension) can be overwhelming. But we do see encouraging results overall (supporting the use of OTCP), and an interest for visualization (e.g. in the taxi dataset, see last picture of https://shorturl.at/alFNM)

---

> > ### Comment · Reviewer_QqAt · 2025-04-07
> >
> > Thanks to the authors for their rebuttal and I appreciate the responses. Below are my comments regarding the authors’ rebuttal:
> >
> > 1. Regarding conditional coverage, first, in conformal prediction, it is well known that conditional validity is impossible to achieve with finite sample size for general distributions; instead, asymptotic conditional coverage is the target. Using binning methods, of course, can achieve conditional validity since only local information is used for constructing the prediction set. But this require the number of bins tend to infinity. I did not get the point why computing the transport map residing in each region would guarantee exchangeability and achieve conditional coverage even with finite sample size. I would appreciate a more detailed and rigorous justification of this claim, as the current argument is unconvincing. Additionally, the binning methods are sensitive to the choice of the number of bins. The rebuttal does not address how this hyperparameter should be selected in practice. Recent work, such as Chernozhukov et al. (2021), has demonstrated that asymptotic conditional coverage can be achieved without binning, suggesting that binning may not be essential for this goal.
> >
> > 2. While I understand the complexity and breadth of datasets used in the experiments, I believe the paper would benefit from deeper insight into the specific types or characteristics of datasets where the proposed method would outperforms existing approaches and give some intuition. Simply showing "encouraging" results or "interesting" visualizations is not sufficient to establish the practical relevance and novelty of the method.
> >
> > Given that the rebuttal does not fully address my concerns, especially with respect to theoretical guarantees and practical guidance, I would increase my score for weak reject.
> >
> > Chernozhukov, Victor, Kaspar Wüthrich, and Yinchu Zhu. "Distributional conformal prediction." Proceedings of the National Academy of Sciences 118.48 (2021): e2107794118.

---

> > > ### Author Response · Authors · 2025-04-09
> > >
> > > We are thankful for your time and valuable comments. We are grateful for your score increase, we will add the clarifications  requested.
> > >
> > > >I did not get the point[...] the current argument is unconvincing."
> > >
> > > Our pipeline (Section 3.2, L 210) boils down to defining a univariate score function, the norm of the composition of a transport map $\hat{T}$ and a multivariate score $S(x,y)$ itself depending on an estimator $\hat{y}(x)$ as $S(x,y)= y - \hat{y}(x)$. In that sense, the fitting of the transport map $\hat{T}$ can be compared to fitting the base model.
> > >
> > > When the model estimation, the clustering procedure and the local transport maps estimation treat the data exchangeably (which is the case), the local conditional validity follows directly from [Proposition 5, Remark 6 and 7](Lei & Wasserman, 2012).
> > >
> > > Naturally this localization cannot recover conditional coverage in finite sample, this would be just an approximation of conditional coverage.
> > >
> > > >Additionally, the binning methods [...] this hyperparameter should be selected in practice.
> > >
> > > We agree, localization creates a trade-off. The partitions should be chosen sufficiently large to contains enough points and small enough to approximate the ground-truth conditional coverage. What we have seen so far for OTCP is:
> > >
> > > * We tried a few “softer” alternatives that consider weighted samples to estimate the OT map, either within clusters or across clusters. We did not see noticeable gains compared to hard `k`-means clustering. Using `k=5,10` gave reasonable results. We agree that setting the number of clusters `k` is an important issue, and one would expect typically `k` to grow with dimension.
> > > * An interesting feature, when using hard clustering, is that if `N` is the total number of score vectors available to estimate the entropic OT map, and `M` the sample size of the uniform ball (e.g. `M=8192` in most of our experiment), then assuming a partition of `N` as `N = N_1 + ... + N_k` into `k` clusters, the compute complexity to recover all `k` Sinkhorn map estimators is not changed, as one would have `O(NM) ~= O(N_1 M) +... +O(N_k M)`.
> > > * When using soft-clustering (i.e. weighted distributions on the `N` points using a kernel), this argument won’t be valid, as each of the `k` problems would run in `O(NM`). One can on the other hand leverage in that case the embarrassingly parallel nature of the Sinkhorn algorithm to compute simultaneously `k` problems `O(NM)` and `k` distributions `a_1, ... , a_k` on the simplex of size `N`.
> > > * Finally, we also tried to rerun (at test time) a reweighted transport (using `K` nearest neighbors) for each new test point. This natural extension was proposed in the context of OT multivariate quantiles in [https://arxiv.org/pdf/2204.11756,](https://arxiv.org/pdf/2204.11756) **Eq. 3.3**). This is far costlier computationally, as it incurs `O(KM)` at each evaluation. This did not change results either.
> > >
> > > >Chernozhukov et al. (2021), has demonstrated that asymptotic conditional coverage can be achieved without binning
> > >
> > > Indeed, one can leverage smoother alternatives, such as those introduced above, using re-weighted transport maps (reweighing is carried out w.r.t. source points, the target points in the uniform ball remain unchanged, as in [https://arxiv.org/pdf/2204.11756,](https://arxiv.org/pdf/2204.11756) (Eq. 3.3).  The quantile regions obtained from this conditional transport map converge asymptotically [Theorem 3.2 and Corollary 3.4](del Bario etal, 2022), hence this would provide in principle asymptotic conditional coverage.
> > >
> > >
> > > >Simply showing "encouraging" results or "interesting" visualizations is not sufficient to establish the practical relevance and novelty of the method.
> > >
> > >
> > > We understand your point.  We still claim, even after adding all of the requested baselines, that OTCP is competitive in moderate dimensions (we drew the line at ≤ 6 in our plots, as mentioned in the beginning of Sec. 4.3).
> > >
> > > Our goal is to turn our visualisation example into a practical tool for spatial prediction with further coding/packaging, We’re confident this can be done and become a reliable contribution (basically any 2D or 3D problem is very well handled, computationally and statistically, with OT, which is why **VQR** was mostly presented for 2D data).
> > >
> > > The performance of localised variants (with no overhead on compute) is more difficult to assess, as this varies with the quality (as you hint) of the clustering process. We plan to split datasets further, beyond dimensionality, to differentiate them w.r.t. low-sample / high-sample problems.
> > >
> > > Finally, we see a recent flurry of activity around flow methods for CP (in 1D so far https://arxiv.org/abs/2406.03346, https://arxiv.org/abs/2502.05709 https://arxiv.org/abs/2406.03346, or recently to appear in ICLR25, in higher dimension https://openreview.net/forum?id=pOO9cqLq7Q ). We believe that on a methodological level only, OTCP is the first to advocate using large scale OT solvers to enrich conformal methods.

---

### Official Review · Reviewer_QpVS · 2025-03-12

**Overall Recommendation:** 4

**Summary:**

The submission proposes to use optimal transport for multivariate conformalized quantile regression. Intuitively, the proposed method first finds the optimal transport map between the unknown data distribution and the uniform ball. Constructing quantile regions in this space is preferable because the problem boils down to conformalizing a scalar value: the distance from the origin. Finally, one can construct quantile regions in the original space by choosing points that have distance from the origin less than the conformalized radius.

Theory is developed to prove distribution-free, finite-sample validity of the procedure, and experiments are carried out on a series of benchmarks. Results show that the proposed method provides confidence region with smaller sizes.

**Claims And Evidence:**

Coverage plots in Fig. 8 are not convincing. Most of the times, the proposed method has average coverage well below the desired level. Coverage is a random variable, so it is not problematic for it to have some failure probability, but on average, it should be around the desired level.

**Essential References Not Discussed:**

There are three essential papers missing from the submission:
[1] Carlier et al. "Vector Quantile Regression: An Optimal Transport Approach"
[2] Feldman et al. "Calibrated Multiple-Output Quantile Regression with Representation Learning", 2023
[3] Rosenberg et al. "Fast Nonlinear Vector Quantile Regression", 2023

[1] Predates Chernozhukov, and develops the optimal transport formulation of vector quantile regression, although not on the unit ball but the unit hypercube
[2] Introduces similar ideas of mapping the data distribution to a centered symmetric distribution where the quantile regions are convex, although by means of a variational autoencoder instead of optimal transport
[3] Uses the ideas of Carlier to develop a scalable method for conformalized vector quantile regression

**Experimental Designs Or Analyses:**

Experiments rely on an existing benchmark for conformal prediction methods

**Methods And Evaluation Criteria:**

The evaluation criteria are appropriate

**Other Comments Or Suggestions:**

**Clarification on dimensionality**

Could the authors clarify the dimensionality of the ball used for optimal transport? This, in the general sense, does not have to be the dimensionality of the multivariate score, correct? What dimensionality is used in practice?

**Confusion about statement on CP**

Lines 225-228 state that conformal prediction does not apply to the "continuous case", could the authors clarify this claim?

**Experiments**

A couple important baselines methods are missing: C-VAE [Feldman et al, 2023], and non linear VQR [Rosenber et al, 2023]. These method also use ideas of optimal transport for multivariate quantile regression. It is important to compare with these methods, both theoretically and empirically.

The coverage plots are only included in the appendix, but they are a fundamental aspect of the contributions. As mentioned above, I was not convinced by the coverage plots, where the expected coverage falls below the required level.

Finally, figures are not cited in the text of the manuscript.

---

**Minor comments**
* The review of Balasubramanian predates most of the contributions mentioned in the paragraph and is likely outdated
* Line 120, right column: "sensitivity error across tasks" is confusing because the text does not define what these tasks are
* Lines 214-219: this paragraph seems to have typos and needs rewriting
* Notation inconsistencies: sometimes norms have their respective $p$ (e.g., $\|\|_2$), sometimes they don't
* Lines 234, right column: typo in $1/2$
* Line 220, right column: Eq. (7) is cited in the text before being written
* Lines 236-263, right column: is the message here that CP works for any function, even those that poorly approximate the map?
* Line 270, right column: repeated "and"
* Proposition 3.4: typos in $\hat{r}_{\alpha, n+1}$ and $\hat{U}_{n+1}$, I assume?
* Line 294: broken crossref

**Other Strengths And Weaknesses:**

**Strenghts**
* The problem of multivariate quantile regression with coverage guarantees is timely
* Optimal transport is a promising technique to solve this issue

**Weaknesses**
* Presentation is rushed, which thwarts clarity
* Missing comparisons with existing methods that use ideas of optimal transport for multivariate quantile regression

My current rating of the paper reflects my doubts on the experimental results, and the missing comparison with existing methods that have explored ideas of optimal transport for multivariate quantile regression. I am looking forward to discussing with the authors!

**Questions For Authors:**

I have no further questions

**Relation To Broader Scientific Literature:**

The submission builds on recent ideas of multivariate quantiles and optimal transport, which are well-established ideas in their respective areas

**Theoretical Claims:**

Propositions 3.4 and 3.5, which are a significant part of the contribution, are presented without proofs.

---

> ### Author Rebuttal · Authors · 2025-04-01
>
> Many thanks for your thoughtful feedback, precise description of shortcomings, and suggestion to add references and baselines.
>
> >**Presentation is rushed, which thwarts clarity**
>
> We apologize for this. We decided to submit based on the publication of a concurrent submission with a similar idea in https://arxiv.org/abs/2501.18991. We have fixed many typos.
>
> >**[1] Predates Chernozhukov [...] [2]**
>
> We agree. Although they were published almost concurrently https://arxiv.org/abs/1406.4643, https://arxiv.org/abs/1412.8434, that important reference was missing. We added it.
>
> > **existing methods that use ideas of optimal transport for multivariate quantile regression**
>
> Thanks for this remark. We wholeheartedly agree that a more detailed discussion on VQR and DQR was missing.
>
> However, we argue that while OTCP and VQR may seem similar, they are very different:
>
> **VQR methods solve a far more challenging problem than OTCP**: they model the conditional quantile map via OT whithin a structural learning problem. While this should *in principle* allow VQR to have a very fine-grained view, and potentially outperform OTCP, we argue that *the reality is messy for $d\geq 2$* and that **aiming** for the most ambitious goal (full view of conditional quantiles) does not necessarily translate into practical gains when used for the simpler goal of uncertainty quantification.
>
> **This difference appears clearly in VQR's computations**. While OTCP uses a standard call to Sinkhorn, VQR requires solving a **mean-independence constrained OT problem**, https://arxiv.org/pdf/2205.14977 [Eq. 4], for which there is no efficient solver other than SGD ("solving VQR via Sinkhorn is very slow in practice"). VQR also requires an extra post-processing step (VMR, Section 5). These practical hurdles, with no consistency guarantees, stand in contrast to our understanding of the Sinkhorn entropic map ([Pooladian/Niles-Weed 21]). Finally, VQR does not inherently provide coverage guarantees unless explicitly combined with conformal techniques with scalar-valued score (e.g. Feldman et al 2023).
>
> **Practically**, the public implementation of (NL)VQR hardcodes the number of target points, so that it grows exponentially in dimension. Please note that https://arxiv.org/pdf/2205.14977 **only consider datasets of dimension $d\leq 2$**.
>
> An important value of our submission, and of the code we have released, is to target upfront scalability issues, which is why we consider higher dimensions (in double digits) in our experiments.
>
> **With all these caveats, we have incorporated the following baselines in our benchmark:**
> * `VQR`, and its conformalized counterpart `VQR-CP`
> * Nonlinear VQR `NL-VQR`, and its conformalized counterpart `NL-VQR-CP`
> * `ST-DQR-CP` [Feldman et al. 23]
>
> **Please look at (anonymized link) https://shorturl.at/alFNM** where we added conditional coverage metrics along with a `localized OTCP`.
>
> Note: the intervals $[0,1/T,.., 1]$ coded in `VQR` was set so that $T^d \approx 8000$, i.e. same # of target points as that used for OTCP.
>
> As can be seen, `OTCP` can perform better than `VQR` methods. These results should not be construed as a criticism of `VQR`. `VQR` aims for a more challenging problem, but may run out of steam in higher dimensions for this specialized task.
>
> >**Propositions 3.4 and 3.5, which are a significant part of the contribution, are presented without proofs.**
>
> Thanks for kindly pointing this out. The proof of Prop. 3.3 requires applying Lemma 3.3 to $Z_i​=S(X_i​,Y_i​$). The proof of Prop. 3.5 is a direct application to the specific case of discrete Spherical uniform. We will clarify.
>
> >**I was not convinced by the  coverage plots [...] expected coverage falls below the required level.**
>
> The coverage can fall below the target level when the exchangeability assumption is not exactly satisfied; Some datasets are very small. We did not attempt to account for robustness.
>
> > “Dimensionality of the ball used for optimal transport?
>
> To ensure the existence of the quantile transport map (inverse of the CDF), the dimensionality of the ball **must be** matched with that of $d$, the multivariate score $S$. There is no other alternative in the literature.
>
> > **conformal prediction does not apply to the "continuous case", could the authors clarify this claim?**
>
> We will remove this. We meant that the mechanics of (continuous) Monge map estimation may collide, at first sight, with the empirical CDF approach that is crucial to CP.
>
> >"is the message here that CP works for any function, even those that poorly approximate the map?”**
>
> A poor map estimation will compromise results in practice, but since we reconformalize the norms of the transports scores, our coverage guarantee trivially holds, just as the coverage does not depend on accuracy of base prediction model. Previous approaches for providing (marginal) coverage for vector-valued map failed see (J. W. Park etal, 2024)[Section 7: Limitations]. See reply to Reviewer **QqAt**

---

> > ### Comment · Reviewer_QpVS · 2025-04-04
> >
> > I sincerely thank the authors for their thoughtful and detailed response to all reviewers' comments and questions.
> >
> > I appreciate the authors' clarification of the differences and connections between OTCP and existing VQR methods, which will be important to include and clarify in the revised version of the paper. I agree with the authors that the toolkit used here is significantly different from existing alternatives, and the contribution is valuable.
> >
> > The extended results with a more comprehensive benchmark are compelling and provide evidence of the claims made in the manuscript. I am happy to raise my score to accept, granted that all promised comparisons and a more clear and thorough discussion of contributions will be included in the revised version of the paper.
> >
> > I am still thinking about the dimensionality of the ball. I agree that in a general setting, the dimension must be $d$ to guarantee existence of the inverse. This might be an advantage compared to VQR, but up to a certain point. I am thinking of very high-dimensional settings (e.g., inverse problems in imaging) where $d$ might be order $10^6$. OTCP might suffer in such cases? However, the score might have a much lower intrinsic dimension (e.g., because pixels are correlated), and one might leverage that, see, for example Belhasin et al [2023], where they use PCA space.

---

> > > ### Author Response · Authors · 2025-04-04
> > >
> > > Dear Reviewer,
> > >
> > > Needless to say that we are extremely grateful for your kind comments and for the strong increase in rating. We are truly thankful for your time reading our rebuttal.
> > >
> > > You can trust our commitment to include all of these baselines (including VQR) in an updated draft. We will maintain the same balanced tone highlighting that VQR aims at a much harder task, and that one should not expect it to perform efficiently for this comparatively easier task.
> > >
> > > > I am still thinking about the dimensionality of the ball. I agree that in a general setting, the dimension must be d to guarantee existence of the inverse. This might be an advantage compared to VQR, but up to a certain point. I am thinking of very high-dimensional settings (e.g., inverse problems in imaging) where d might be order 10^6. OTCP might suffer in such cases? However, the score might have a much lower intrinsic dimension (e.g., because pixels are correlated), and one might leverage that, see, for example Belhasin et al [2023], where they use PCA space.
> > >
> > >
> > > This is an excellent point, and thanks for the great reference that we will be happy to include. At this point, we can only speculate on practical approaches to deal with high dimensional scores.
> > >
> > > As you hint with your reference to **[Belhasin et al. 2023]**, a foolproof solution would be to capture a low intrinsic-dimension for high-dimensional score vectors, carry out efficiently that dimensionality reduction, and transport these scores to the ball of lower dimension. For instance, if the scores were mapped using a VAE encoder/decoder pair $(e,g)$ (which would need to be trained on held-out data to guarantee coverage), one could still maintain some loose form of invertibility (using the decoder) and still recover samples of the ball in the original space if needed.
> > >
> > > So for instance, writing $T$ from the OT map from the measure of encoded scores $e(S(X,Y))$ to the $d$-ball, the conformity of an input/output pair would be assessed as $\|\|T(e(S(x,y)))\|\|_2$, while samples could be generated by sampling $z$ from the uniform ball in dimension $d$ to generate $ \hat{y}-g\circ T^{-1}(z)$.
> > >
> > > Of course, a far more ambitious solution would be to learn **directly** the transport from the space of score vectors to a lower dimensional uniform reference ball. The technical difficulty in this case is to define an appropriate cost $c(s,z)$, $c:\mathbb{R}^p\times \mathbb{R}^d\rightarrow \mathbb{R}$ between these two spaces. This is usually handled through variants of quadratic optimal transport (i.e. Gromov-Wasserstein), but the existence of such maps is very much an open problem (https://proceedings.mlr.press/v238/sebbouh24a.html, https://arxiv.org/abs/1806.09277, https://arxiv.org/abs/2210.11945) and this problem is likely much harder. If one were to follow that route, https://proceedings.mlr.press/v238/sebbouh24a.html proposes a procedure to generalize the entropic map so that it works across dimensions.
> > >
> > > With our sincere gratitude for your update,
> > > The Authors

---

### Official Review · Reviewer_8VoF · 2025-03-14

**Overall Recommendation:** 4

**Summary:**

This paper introduces a conformal prediction method that constructs quantile regions for multivariate conformity scores using optimal transport. The authors provide finite-sample guarantees for both the exact optimal transport map and its more computationally efficient approximations.

**Claims And Evidence:**

Yes.

**Essential References Not Discussed:**

To my knowledge, all relevant related work is cited.

**Experimental Designs Or Analyses:**

I find the empirical results not entirely convincing. The computation time can be significantly higher for only a slight reduction in the size of the prediction set compared to other methods. Additionally, some key metrics are missing, such as conditional coverage for Figures 1 and 4, which would help in properly interpreting the results.

For Figures 2 and 8, it is not evident that the size of the predicted regions decreases as *m* increases. Moreover, Merge-CP appears to perform better on certain datasets (e.g., Figure 4: datasets *wq, oes10, oes97,* and *scm1d,* as well as Figure 9). It would be useful to explain why this occurs and under which conditions OT-CP outperforms other methods. Additionally, is the conditional coverage of OT-CP superior to other approaches?

Figure 5 is not clear, and the legend in Figure 9 should be repositioned for better readability.

The baselines M-CP, Merge-CP and Merge-CP (Mah) are quite simple, but more advanced methods generally require access to a generative model. While no precise hyperparameter tuning is performed for $m$ and $\epsilon$, I agree that reasonable defaults are sufficient in this case. These hyperparameters are compared in Figures 2, 8 and 10. However, I don't observe a significant difference between the figures and no interpretation is proposed.

**Methods And Evaluation Criteria:**

The method ensures finite-sample coverage even when approximating the optimal transport map, enhancing robustness. It is evaluated on 24 benchmark datasets from a previous study, reinforcing the validity of the results. The region size and conditional coverage are relevant metrics; however, conditional coverage metrics are not explicitly evaluated despite their importance.

**Other Comments Or Suggestions:**

- How do the authors generate n_S vectors uniformly on the unit sphere in dimension d?
- Page 6: A question mark remains in the second column.
- Page 4: A capital letter appears instead of a lowercase one in the sentence: "When dealing with empirical distribution with finite samples Z1,...,Zn,Zn+1in this asymptotic regime,..."

**Other Strengths And Weaknesses:**

### *Strengths:*
- Exploring quantile regions for multivariate scores is an important research direction.
- The proposed framework is general and applicable to any multivariate conformity score, making it flexible and widely usable.

### *Weaknesses:*
- The paper focuses exclusively on multi-output conformal methods that do not require estimating the joint distribution of $ Y_{n=1} $, as in [1] and [2]. However, this restriction is not explicitly justified. In particular, it is unclear whether methods that estimate the joint distribution would necessarily incur higher computational costs than those relying on exact or approximate optimal transport maps.
- While the vector-valued conformity score is conditioned on \( x \), the optimal transport map itself is not, raising concerns about the method’s ability to fully capture conditional uncertainty. Notably, [3] extended Hallin et al. (2021) to conditional quantile regions, which may offer a more flexible alternative.
- Some figures, such as Figures 2 and 3, are difficult to read and should be improved.
- Illustrations of the prediction regions would have been helpful in better understanding what OT-CP does, and providing concrete examples could have better motivated the proposed method.
- The lack of publicly available code limits reproducibility.


[1] Feldman, Shai et al. “Calibrated Multiple-Output Quantile Regression with Representation Learning.” JMLR (2023).
[2] Wang, Zhendong et al “Probabilistic Conformal Prediction Using Conditional Random Samples.” In AISTATS 2023.
[3] del Barrio et al. 2024. “Nonparametric Multiple-Output Center-Outward Quantile Regression.” Journal of the American Statistical Association.

**Questions For Authors:**

Please see above.

**Relation To Broader Scientific Literature:**

This paper lies at the intersection of optimal transport and conformal prediction, both of which are active research areas. In optimal transport, a particularly relevant recent contribution is the work of Hallin et al. (2021), which introduced quantile regions by ordering vectors based on optimal transport. This idea of leveraging optimal transport for distributional inference aligns with the methodological foundation of the present work.

In conformal prediction, several recent methods have been proposed, including those by Izbicki et al. (2022), Wang et al. (2022), and Dheur et al. (2024). These approaches typically assume the availability of a generative model, which facilitates the construction of prediction sets. In contrast, the present paper does not rely on this assumption, which constrains the selection of baseline methods.

**Theoretical Claims:**

The different theoretical propositions are correct, and the only proof provided is also correct.

---

> ### Author Rebuttal · Authors · 2025-04-01
>
> Many thanks for your detailed and insightful review.
>
> > **conditional coverage are relevant metrics**
>
> We agree and now include conditional coverage metrics. We also implement a localized variant of `OTCP` where the data is partitioned in the feature space using $k$-means ($k=5$ or $10$) and a separate transport map is learned in each region. This offers a simple way to capture conditional heterogeneity.
>
> **Additional experiments with conditional coverage metrics as suggested can be found in https://shorturl.at/alFNM**
>
> As expected, the localized version can indeed improve the worst-case conditional coverage but with additional computations to fit transport map on every partitions.
>
> > **The computation time can be significantly higher for only a slight reduction in the size of the prediction set compared to other methods. […]For Figures 2 and 8, it is not evident that the size of the predicted regions decreases as m increases. Moreover, Merge-CP appears to perform better on certain datasets**
>
> This is a fair observation, as the computation of a transport map is more costly. However, the gain in region size can be substantial. We do not claim that `OTCP` outperforms all methods in all datasets. From our observations, higher dimensionality can degrade the quality of multivariate quantiles L.260. We can still guarantee that coverage is provably maintained but the size of the region can indeed be inflated depending on the approximation errors.
>
> > **"The paper focuses exclusively on multi-output conformal methods that do not require estimating the joint distribution”**
>
> This is a valid comment. If one has access to the an estimate of the joint distribution, it should be interesting to see how to wrap it with the `OTCP` framework. In this paper, we focused on clarifying the situation in a model-agnostic and arbitrary score. If one has access to a  smooth joint or conditional distribution $P_{Y \mid X}$ One natural strategy is to follow PCP style for examples and consider vector-valued scores on samples obtained from the generative model.
>
> Additionally, one could also extract a conditional transport map $T_x \\# P_{Y \mid X=x} = \mathbb{U}$ to a reference distribution. Our framework can still operate as a wrapper around generative models by being compatible with any estimated (conditional) transport map on the output.  Indeed if a variable $Z$ can be transported by $T$, then an invertible function $f$ of $Z$ also induces a transport map $T_f = f\circ T \circ f^{-1}$. As such, a transport map on $Y$ also induces a natural transport map by composition on the conformity score $s(x,y)$ simply by applying this to  $f= s(x, \cdot)$ for each $x$. As such, we can easily incorporate it in OTCP pipeline to provide valid coverage. We will add more clarification on this in the revised version.
>
> > **While the vector-valued conformity score is conditioned on ( x ), the optimal  transport map itself is not, raising concerns about the method’s ability to fully capture conditional uncertainty.**
>
> This is a good remark and corresponds to exactly what we implemented as `localized OTCP` with a $k$-means strategy. Basically each point will have its own transportation map and we can think of the OT merging score as $S_{\mathrm{OTCP}}^{A_k}(x, y) = \|T_{A_k} \circ S(x, y)\| $
> where the collection of $(A_k)_{k\in[K]}$ is a clustering partition of the feature space. This matches the suggested approach leveraging conditional transport map in Hallin et al. (2021)
>
> > **“How do the authors generate $n_S$ vectors uniformly on the unit sphere in dimension $d$?”**
>
> We formally describe it in the paper L.324 *"Sampling on the sphere"*. We use a quasi-Monte Carlo method to generate points evenly spread on a sphere. Starting from well-distributed points in $[0,1]^d$, we transform them using the inverse normal distribution to get Gaussian-like vectors, then normalize them to lie on the unit sphere. This gives us a lower-discrepancy sampling of directions than random sampling.
>
> > **Illustrations of the prediction regions would have been helpful in better understanding what OT-CP does, and providing concrete examples could have better motivated the proposed method.**
>
> We provided the taxi demand prediction task and included a new visualization with the localized transport maps.
> Thanks for the suggestions for improving our figures.
>
> > **“The lack of publicly available code limits reproducibility.”**
>
> We have already released an `OTCP` module in a major optimal transport toolbox, that can be used to directly recover our experimental results. To preserve anonymity, we cannot include the link now but it will be included in the camera ready version of our paper.

---

> > ### Comment · Reviewer_8VoF · 2025-04-06
> >
> > We thank the authors for their response. We have increased our score.

---

### Decision · Program_Chairs · 2025-05-01

**Decision:**

Reject

**Comment:**

This paper introduces a conformal prediction method that constructs quantile regions for multivariate conformity scores using optimal transport. The authors provide finite-sample guarantees for both the exact optimal transport map and its more computationally efficient approximations.

The problem of multivariate quantile regression with coverage guarantees is timely and optimal transport is a promising technique to solve this issue. The initial version has many weaknesses as reflected in the original scores and comments.  However, the authors managed to convinced all reviewers to raise their scores, some substantially. Yet, one reviewer still has concerns about the theoretical guarantees and practical guidance, and thinks the theory should covers conditional coverage.  I agree with this view.